# Unraveling the Radioprotective Mechanisms of UV-Resistant *Bacillus subtilis* ASM-1 Extracted Compounds through Molecular Docking

**DOI:** 10.3390/ph16081139

**Published:** 2023-08-11

**Authors:** Asim Ur Rahman, Aftab Ali, Faisal Ahmad, Sajjad Ahmad, Metab Alharbi, Abdullah F. Alasmari, Amna Fayyaz, Qurrat ul ain Rana, Samiullah Khan, Fariha Hasan, Malik Badshah, Aamer Ali Shah

**Affiliations:** 1Department of Microbiology, Quaid-i-Azam University, Islamabad 45320, Pakistan; arahman@bs.qau.edu.pk (A.U.R.); aftabkhan21995@gmail.com (A.A.); samikhan@qau.edu.pk (S.K.); farihahasan@yahoo.com (F.H.); malikbadshah@qau.edu.pk (M.B.); 2National Center for Bioinformatics, Quaid-i-Azam University, Islamabad 45320, Pakistan; faisalahmad@bs.qau.edu.pk; 3Department of Health and Biological Sciences, Abasyn University, Peshawar 25000, Pakistan; sajjad.ahmad@abasyn.edu.pk; 4Gilbert and Rose-Marie Chagoury School of Medicine, Lebanese American University, Beirut P.O. Box 36, Lebanon; 5Department of Natural Sciences, Lebanese American University, Beirut P.O. Box 36, Lebanon; 6Department of Pharmacology and Toxicology, College of Pharmacy, King Saud University, P.O. Box 2455, Riyadh 11451, Saudi Arabia; mesalharbi@ksu.edu.sa (M.A.); afalasmari@ksu.edu.sa (A.F.A.); 7Department of Environmental Sciences, Quaid-i-Azam University, Islamabad 45320, Pakistan; amnafayyaz505@gmail.com; 8Joint Genome Institute, Lawrence Berkely National Laboratory, Berkley, CA 94720, USA

**Keywords:** antioxidants, radioresistant microorganisms, radioprotection, biofilm, co-operative growth, oxidative stress

## Abstract

Radioresistant microorganisms possess inimitable capabilities enabling them to thrive under extreme radiation. However, the existence of radiosensitive microorganisms inhabiting such an inhospitable environment is still a mystery. The current study examines the potential of radioresistant microorganisms to protect radiosensitive microorganisms in harsh environments. *Bacillus subtilis* strain ASM-1 was isolated from the Thal desert in Pakistan and evaluated for antioxidative and radioprotective potential after being exposed to UV radiation. The strain exhibited 54.91% survivability under UVB radiation (5.424 × 10^3^ J/m^2^ for 8 min) and 50.94% to mitomycin-C (4 µg/mL). Extracellular fractions collected from ASM-1 extracts showed significant antioxidant potential, and chemical profiling revealed a pool of bioactive compounds, including pyrrolopyrazines, amides, alcoholics, and phenolics. The E-2 fraction showed the maximum antioxidant potential via DPPH assay (75%), and H_2_O_2_ scavenging assay (68%). A combination of ASM-1 supernatant with E-2 fraction (50 µL in a ratio of 2:1) provided substantial protection to radiosensitive cell types, *Bacillus altitudinis* ASM-9 (MT722073) and *E. coli* (ATCC 10536), under UVB radiation. Docking studies reveal that the compound supported by literature against the target proteins have strong binding affinities which further inferred its medical uses in health care treatment. This is followed by molecular dynamic simulations where it was observed among trajectories that there were no significant changes in major secondary structure elements, despite the presence of naturally flexible loops. This behavior can be interpreted as a strategy to enhance intermolecular conformational stability as the simulation progresses. Thus, our study concludes that *Bacillus subtilis* ASM-1 protects radiosensitive strains from radiation-induced injuries via biofilm formation and secretion of antioxidative and radioprotective compounds in the environment.

## 1. Introduction

Microorganisms that survive in extreme environmental conditions, like high desiccation and prolonged exposure to radiation, exhibit unique survival strategies which enable them to grow optimally in such life-threatening conditions [1]. Radiation triggers DNA damage and induces oxidative injury to vital biomolecules through the radiolysis of water [2,3]. In response to these detrimental effects of radiation, radioresistant microorganisms developed advanced genome repair mechanisms accompanied by various radiation-responsive secondary metabolic products such as extremolytes, including photoprotective pigments, that can absorb radiation [4,5]. In addition to these, the enzymatic and non-enzymatic antioxidant systems also assist in the survival of these microorganisms by neutralizing and scavenging reactive radicals and other unstable compounds which are produced due to radiation exposure [6].

Microbial bio-pigments that have an array of targeted biological properties are now being evaluated by researchers worldwide against numerous human diseases in the form of skin care agents, immunosuppressants, antioxidants, and anti-tumor agents, and are, therefore, considered potential candidates for the pharmaceutical and cosmetic industries. So far, the extremolytes utilized as radioprotectants in skin care utilities are sioxanthin, astaxanthin, shinorine, scytonemine, pannarin, melanin, biopterin, bacterioruberin, and microsporine-like amino acids [7]. In a nutshell, radioresistant extremophiles act as a reservoir for novel bioactive compounds that are not only important for their structural and biochemical diversity but also have extensive biotechnological applications.

Besides this, there are a vast number of scientific reports available on the metabolites-aided survival capabilities of radioresistant microorganisms inhabiting such an inhospitable environment. However, in the process of sampling, isolation, and screening, you will also obtain a few radiosensitive microorganisms, which raises a question of concern: how are radiosensitive microorganisms able to survive in such hostile environments? In one of the previous reports, Igor et al. studied the co-operative growth between radioresistant and radiosensitive microbial cells inhabiting the same ecological settings, using a genetic approach targeting only catalase enzymes (by creating mutants) [8]. However, the actual survival mechanism of such radiosensitive microorganisms is still unclear and poses a new era of research, investigating the symbiotic relationships between microorganisms living in extreme environments.

In this connection, the present study evaluates the in vitro biological activities and characterization of bioactive metabolites from a radioresistant bacterium, *Bacillus subtilis* ASM-1 isolated from Thal desert of Pakistan. Furthermore, the radioprotective properties of the extracted bioactive compounds isolated from strain ASM-1 were also assessed using a new approach, i.e., protecting radiosensitive microbial strains *Bacillus altitudinis* ASM-9 and *E. coli* (ATCC 10536) from radiation-induced damage.

## 2. Results

### 2.1. Isolation and Screening for Radioresistant Bacteria

A total of 33 bacterial isolates recovered from Chashma (TMC) and the Makarwal region (ASM) were subjected to UV radiation to determine their behavior. Among 33 isolates, 18 isolates from Chashma and 8 from the Makarwal region were found to be resistant to UV radiation in primary screening for 5 min. In secondary screening (radiation dose for more than 5 min), the UVB dosage to all resistant isolates was gradually increased by increasing exposure time from 3 to 20 min. Strain ASM-1 was selected based on resistance to a maximum energy dose, as shown in Table 1.

### 2.2. Characterization of Strain ASM-1

Strain ASM-1 was observed to be Gram-positive, growing in dry, opaque, circular colonies with dark red colored pigmentation on TGY agar plates. The strain was positive for amylase, catalase, superoxide dismutase, peroxidase, cellulase, and lipase enzymes, while negative for protease enzyme. A maximum similarity of 100% was shown against *Bacillus* sp. when a 16S rRNA gene sequence of our potential strains was subjected to BLAST analysis. This was identified as *Bacillus subtilis* strain ASM-1. A phylogenetic tree was constructed using the Neighbor-Joining method to deduce the evolutionary relationship of ASM-1 with other strains already reported in NCBI GenBank (Figure 1). The nucleotide sequence was submitted to NCBI GenBank with accession number (OK559666).

### 2.3. Survival Rate of Bacillus subtilis Strain ASM-1 under UVB Radiation and Mitomycin-C

*B. subtilis* strain ASM-1 exhibited 54.91% survivability against a control strain *E. coli* (ATCC 10536) that showed 20.23% under the same dose of UVB radiation, i.e., 5.424 × 10^3^ J/m^2^ (8 min). Likewise, the control strain showed no growth as compared to ASM-1 under UVB dose of 8.136 × 10^3^ J/m^2^ (12 min), as shown in Figure 2A. Similarly, ASM-1 showed 50.90% survivability at 4 µg/mL under Mitomycin-C exposure as compared to a control *E. coli* strain which showed only 25.63%, respectively (Figure 2B).

### 2.4. Purification and Selection of the Extracellular Crude Extracts

The extracellular crude extracts (AS-2) resulted in four fractions in their corresponding solvents. All fractions were primarily evaluated for their antioxidant capability via DPPH assay. Fractions E-2 and D-2 possessed maximum antioxidant activity, while the remaining fractions W-2 and M-2 exhibited insignificant activities. The fractions E-2 and D-2 were suspended in their respective solvents and stored at 4 °C for bioassays and analytical purposes.

### 2.5. In Vitro Evaluation for the Antioxidant Potential of the Selected Fractions

#### 2.5.1. DPPH Assay

Results demonstrated that scavenging activity of the fractions E-2 and D-2 was increased with their increasing concentrations. The E-2 fraction accounted for 75.55% of DPPH free radical scavenging potential as compared to ascorbic acid (ASA) as a positive control, which showed 80.43% activity at a concentration of 250 µg/mL, while at this concentration, the D-2 fraction exhibited the lowest scavenging potential of 60.37% (Figure 3A).

#### 2.5.2. Hydrogen Peroxide Scavenging Assay

Hydrogen peroxide scavenging was also observed in a concentration-dependent manner. However, the E-2 fraction showed 68.53% scavenging potential in comparison to the D-2 fraction (63.37%) against a positive control ASA that provided 66.84% scavenging at a concentration of 250 µg/mL (Figure 3B).

#### 2.5.3. Cytotoxic and Antibacterial Potential

The cytotoxic effect on brine shrimp larvae was perceived only for the D-2 fraction as compared to the E-2 fraction. An approximately 61% killing effect was detected for D-2 as compared to 74% for Mitomycin-C at the maximum concentration of 250 µg/mL. All fractions exhibited insignificant antibacterial activities, except for the D-2 fraction that possessed a slight antibacterial effect against a clinical strain of *Staphylococcus aureus* with ZIB of (7 mm). However, no fraction was found to have antibacterial activity against ASM-9 and *E. coli* (test strains).

#### 2.5.4. Biofilm Detection Assay

The in vitro evaluation for biofilm formation ability was tested for both isolates, i.e., ASM-1, and ASM-9 at three-time intervals (24, 48, and 72 h). Both isolates showed biofilm formation potential in a diminishing manner. ASM-1 showed maximum biofilm potential as compared to ASM-9 after 24 h (Figure 3C).

#### 2.5.5. GC–MS Analysis of the Selected Fractions

The selected ASM-1 fractions (E-2 and D-2) which exhibited maximum antioxidant activities were subjected to their chemical profiling through GC/MS analysis. The GC/MS spectra unveiled a pool of volatile compounds that were carefully identified based on their spectral masses upon comparing them to the online National Institute of Standards and Technology (NIST) databases. In the present study, a pool of chemical compounds from different classes in all perceived fractions was Pyrrolopyrazine, Alkane, Phenol, Dicarboxylic acid butyl ester, Dicarboxylic acid octyl ester, Fatty acid, Amide, β-carboline alkaloid and Alcohol. Dicarboxylic acid butyl ester was the only compound detected at RT 21.83 in fraction D-2. The detailed information of the chemical compounds including their names, retention time, molecular weight, molecular formula, and class are listed in Table 2.

### 2.6. Radioprotective Activity of ASM-1 Extracts on Selected Radiosensitive Strains

*Bacillus subtilis* strain ASM-1 extracts had been assessed for their potential to facilitate radioprotective effect on selected radiosensitive bacterial strains ASM-9 and *E. coli* under different doses of UVB radiation. For this purpose, the E-2 fraction was selected against the D-2 fraction because of its significant antioxidant potential and having no cytotoxic and antibacterial capacity. The survivability of test strains was reduced under UVB radiation in a dose-dependent manner. With no extracts added, *E. coli* (ATCC 10536) exhibited 19.69% survivability, while strain ASM-9 showed 0.33% survivability under an energy dose of 3.390 × 10^3^ J/m^2^ (5 min) (Figure 4A). Their survivability was significantly enhanced from (19.69% to 44.92%) for *E. coli*, and for ASM-9 survivability was enhanced from (0.33% to 21.62%) after the addition of the mixture (E-2 fraction and supernatant) under an acute UVB dose of 3.390 × 10^3^ J/m^2^ for 5 min (Figure 4B).

### 2.7. Ligand Preparations and Molecular Docking

Molecular docking of the compounds was carried out using the PyRx interface of AutoDock Vina and GOLD software [9]. For the docking process, the following parameters were employed: an exhaustive value of 300, the binding site was defined as the residues of the N-terminal domain involved in the binding pocket, and a maximum of 10 poses were allowed. The PyRx tool generated binding affinity values as negative values (where a more negative value indicates a stronger binding affinity). The docking results of the top inhibitors can be found in Figure 5 and Figure 6. To validate the docking protocol used, extracted compounds were docked into the N-terminal structure of (PDB ID = 5F1A and 5ZKP). The binding mode as the one observed in crystallization studies for the known inhibitors was obtained. The docking reproducibility results obtained using AutoDock Vina and GOLD are presented in Figure 5 and Figure 6.

Protein–ligand docking is an important technique for accurately predicting the orientation of a ligand within its target protein [10]. Compounds that exhibited strong binding affinity and correct binding poses were selected. The binding affinity represents the total torsional energy, internal energy, and intermolecular energy subtracted from the unbound energy system. A high binding affinity indicates a stable protein–ligand complex. Interestingly, the compounds docked in similar conformations to both the full-length 5F1A structure and the N-terminal structure alone. As the C-terminal region did not significantly contribute to the binding of compounds at the N-terminal, this study focused on the N-terminal structure alone. To determine the binding poses, interactions, and binding affinity of both the inhibitors, they were docked into the binding pocket of 5F1A and 5ZKP using the PyRx tool and GOLD software. Each inhibitor generated 10 poses, and those with strong binding affinity were chosen (Figure 5 and Figure 6). The docking results revealed a range of binding affinities from −7.6 to −7.1 kcal/mol with Goldscore of 56 and 53 against 5F1A, whereas 53 to 51 against 5ZKP for compounds **1** and **2**. The analysis showed that compound **1** and compound **2** shared a motif that bound to the same residues (Thr175, His176, Gln172, Trp356, Ala168, Leu359, Ala171, Tyr354, His357, His355, Asn351, Phe155). Moreover, a hydrogen bond interaction with the active site residues (His357 and Thr175) was observed for the shared motif in compound **1** and compound **2**, whereas against protein target 5ZKP, the shared active residues involved during binding interaction are Tyr17, Tyr72, Leu400, Trp68, Ala88, Gly89, Phe92, Phe155, His396, Gln397, Arg9, Glu156, Cys154 and Tyr71, while Tyr17, Tyr72 and Glu are involved in hydrogen bond formations in both compounds **1** and **2** in Figure 5 and Figure 6.

### 2.8. Molecular Dynamics Simulation Analysis

To validate the docking, the findings and validate intermolecular stable conformation between the enzymes and compounds, molecular dynamics simulation analysis was conducted. The simulation was carried out for 20 ns. The simulation trajectories were used root mean square deviation (RMSD) analysis to shed light on enzyme carbon alpha atoms deviations during simulation time. As can be seen in Figure 7, all four docked complexes revealed stable dynamics with no drastic local or global changes observed. The 5f1A-a-56 system was noticed as the most stable with a mean RMSD value of 0.51 Å. The mean RMSD of 5faA-2-53, 5zkp2-51 and 5zkp1-51 was 0.89 Å, 1.32 Å, and 1.33 Å, respectively (Figure 7). Upon trajectory visualizations, no major secondary structure elements changes were witnessed despite naturally flexible loops, which may be explained as an approach to provide more intermolecular conformational stability as the simulation time proceeds.

## 3. Discussion

Actinomycetes, followed by Bacillus, are considered a reservoir for their numerous vital metabolites including antibacterials, antifungals, antiparasitics, antioxidants, growth-promoting substances, immune modifiers, immunosuppressants, and enzyme inhibitors [11]. These natural products have bioactive properties that have been extracted from microorganisms for decades to serve as potential candidates for a variety of industrial sectors [12]. As a result, there is always a need for novel and highly potent metabolites that can be obtained from previously untapped extremophilic microbial sources [1,13]. Similarly, understanding the effects of acute and chronic radiation on radiosensitive and radioresistant microbial cell types is a potentially important new tool for understanding the symbiotic relationships among microorganisms inhabiting such hostile environments [7].

*Bacillus subtilis* strain ASM-1, a radioresistant bacterium, was isolated from an unexplored environment in Pakistan’s Thal desert. *Bacillus* sp. are thought to be one of the most common inhabitants in such hostile environments due to their endospore and biofilm formation abilities, as well as the secretion of several vital metabolites. Similarly, the isolate’s production of enzymes, such as catalases and peroxidases, demonstrated the strain’s enzymatic antioxidant potential to counterbalance the oxidative stress produced intracellularly and extracellularly by multiple oxidants [11].

The survival efficiency of the ASM-1 strain was investigated by subjecting it to various physical and chemical stressors, i.e., UVB radiation and Mitomycin-C. UVB radiation can cause oxidative stress or impose the formation of thiamine dimers, which are difficult to transcribe and repair, ultimately resulting in cell death. Mitomycin-C, on the other hand, is an anticancer agent that inhibits abnormal cell proliferation. Studies revealed that ASM-1 could withstand the damaging effects of these stressors up to a certain level. This is most likely due to the fact that radioresistant microorganisms evolved multiple defense mechanisms, such as the production of UV-absorbing compounds, enzymatic and non-enzymatic antioxidant systems, and an efficient DNA repair system, allowing them to survive in such harsh environments [14]. The ASM-1 strain was chosen for further investigation due to its elevated resistance to both stressors.

Radiation-induced oxidative stress is harmful to all life forms, so compounds that counteract this oxidative stress are recognized as radioprotective agents. These compounds’ primary methods of radioprotection include cellular oxygen tension decrease, free radical scavenging, and hydrogen transfer [15]. This study used two antioxidant assays to assess the potency of ASM-1 crude extracts as an antioxidative and radioprotective agent against radiation-induced oxidative stress. The assays chosen were solely based on the extracts’ potential future implications (investigating their potential role to aid in the survival of radiosensitive test strains under radiation-induced oxidative stress). ASM-1 extracts demonstrated significant antioxidant potential in a dose-dependent manner by scavenging DPPH free radicals and hydrogen peroxide species [16,17]. The use of multiple bioassays revealed that ASM-1 extracts had significant bioactivity against multiple reactive oxygen species (ROS). These findings imply that ASM-1 extracts may be useful in preventing ROS-mediated damage.

Both fractions were promoted for further bioassays, i.e., cytotoxic, antibacterial, and biofilm formation potential, to achieve the study’s intended goal. As the E-2 fraction had no significant antibacterial and cytotoxic potential compared to the D-2 fraction, which had little cytotoxic and antibacterial potential, only the E-2 fraction was preferred for the radioprotective experiment. Similarly, the biofilm formation potential of both strains was observed (ASM-1 and ASM-9). However, ASM-1 demonstrated significantly greater biofilm formation ability than ASM-9 after 24 h, which then declined after 72 h of incubation [18].

Similarly, ASM-1 fractions, particularly E-2 and D-2, were chemically characterized by the employment of GC–MS analysis. The GC–MS analysis for the ASM-1 fractions revealed several chemical compounds including alcohols, phenols, pyrrolopyrazine, alkaloids, amides, and different types of acids. Some of the chemical constituents of the GC–MS spectra have been previously reported to have antioxidative, cytoprotective, and radioprotective capacities. In the present study, the E-2 fraction comprised of two pyrrolopyrazine compounds, Pyrrolo [1,2-a] pyrazine -1,4-dione, hexahydro-3-(2- methyl propyl) and Pyrrolo [1,2a] pyrazine-1,4-dione, hexahydro -3-(phenylmethyl), that are well-known compounds to have antioxidative and radioprotective potential, as previously described [19]. Similarly, 9H-Pyrido [3,4-b] indole is a tricyclic indole β-carboline alkaloid that inhibits radiation-induced oxidation transformation reactions [20]. Dodecane is another long-chain alkane detected in GC–MS spectra to have a significant antioxidant potential [21]. Likewise, n-Nonadecanol-1 is long-chain alcohol and 1, 2-benzene dicarboxylic acid that possesses antimicrobial and cytotoxic properties, respectively [22]. Likewise, the D-2 fractions were comprised of 1-Octanol, previously detected in *Bacillus subtilis* CF-3 extracts to have bioactive potential, While several benzoic acids in the extracts have already been reported in *Bacillus* sp. to have biological activities [23], benzamide was the only compound possessing anticancer and other biological activities. The chemical constituents detected in the GC/MS spectra of ASM-1 fractions suggested the antioxidant potential of *Bacillus subtilis* strain ASM-1. Therefore, this study provides a shred of evidence on the antioxidative and radioprotective role of the metabolites derived from *Bacillus subtilis* ASM-1.

Several tests were performed to determine the potential of ASM-1 extracts against oxidative stress, bacterial competitors, and cytotoxicity. These findings assumed the E-2 fraction’s safe and intended effect in the radioprotective experiment. In a radioprotective experiment, administering only the E-2 fraction did not result in a significant change. Therefore, ASM-1 supernatant was also considered because it contains antioxidant enzymes, UV-absorbing compounds, and exopolysaccharides (a prerequisite for biofilm development). ASM-1 supernatant from a 24-hour-old culture was collected for this purpose because significant biofilm formation was detected at this stage. As a result, a recipe (Supernatant and E-2 fraction in 2:1) was developed after several trials to obtain enhanced survivability. ASM-1 was found to have significant radioprotective activity, meaning that the extracellular extracts protected radiosensitive *E. coli* and ASM-9 from the harmful effects of UVB rays far beyond the standard dose (5 min). The reason for this improved survivability could be that the E-2 fraction provided the necessary components (antioxidants), whereas the supernatant provided antioxidative enzymes, UV-absorbing compounds, and exopolysaccharides. Furthermore, after the irradiation dose, well-defined clumps of ASM-9 cells were detected on TGY agar plates, which could be due to the exopolysaccharides that glued the ASM-9 cells for enhanced survival. However, the precise mechanism is unknown and requires further exploration. The findings were like those of Igor et al., who used a molecular model to investigate the radioprotective and antioxidative potential of radioresistant *Dinococcus radiodurans* on radiosensitive *E. coli*, specifically silencing the catalase enzyme gene [8]. This experimental setup, however, was modified to investigate the radioprotective effect of radioresistant ASM-1 extracts on radiosensitive strains in the environment. Furthermore, some of the compounds obtained were searched out against the UV protections where two compounds were identified as reported to be used against UV radiation in sunscreen. These compounds includes Pyrrolo [1,2-a]pyrazine-1,4-dione, hexahydro-3-(2-methylpropyl) and 9H-Pyrido [3,4-b]indole. Both compounds were docked against the reported protein targets which is responsible for involved in UV-induced photodamage, such as serotonin-receptor subtype 5-HT2A, platelet-activating factor receptor (5ZKP), and Aspirin and other nonsteroidal anti-inflammatory drugs target the cyclooxygenase enzymes (COX-1 and COX-2) to block the formation of prostaglandins (5F1A). Molecular docking inferred potential binding affinities of the hit compounds against both the protein targets which open a way towards further investigations in future. Further these docking results have been validated via molecular dynamic simulations which emphasize the confirmational behavior in a real time environment. Interestingly, it has been deduced that no confirmational changes were recorded after 20 ns of simulation time interval against all the complex systems, despite some loop changes and the ligand remaining in contact with active residues of the active pocket.

These results shed light on the symbiotic relationships between radioresistant and radiosensitive microbial cells that co-exist in the same environmental niche exposed to chronic radiation. The radioresistant microorganisms’ indefinite defense capabilities might aid the survival means for radiosensitive strains by secreting numerous potent metabolites, and by forming a protective barn (biofilm) for the same and different microbial species in such an unrelenting environment with extreme oxidative stress caused by radiation, as per the first-time study in this regard [24]. In a symbiotic relationship, radioresistant microbial species could allow radiosensitive microbial cells (effective degraders) to detoxify radioactive waste sites. Similarly, metabolites derived from radioresistant microorganisms can protect humans and other life forms from radiation-induced toxicity, particularly carcinogenesis.

## 4. Materials and Methods

### 4.1. Isolation and Screening of Radioresistant Bacteria

Soil samples were aseptically collected from the surface and subsurface (2 cm depth) using standard microbiological protocol from two different locations., i.e., the Chashma and Makarwal regions of the Thal desert, Pakistan. The samples were serially diluted in double distilled water, followed by inoculation on Tryptone Glucose Yeast (TGY) medium by spread plate method. The composition of TGY medium was (g/L): tryptone 10, yeast extract 5; glucose 1. The TGY plates were exposed to UV radiation for 3 and 5 min in the UV chamber (119 cm × 69 cm × 52 cm) supplied with a 20 W, 280 nm UV light before incubation. The test sample’s UV fluence rate was determined using the following equation [25].
He = Ee × t(1)
where He is the energy that reaches a surface area due to irradiance (Ee) maintained for the time duration (t). The UV fluence rate is defined as the energy of radiation that reaches a surface area in a definite period (energy/area/time) measured in J/m^2^. Exposure duration to the UV fluence rate was used to compute the total UV dosage. Subsequently, plates were wrapped in aluminum foil and incubated at 37 °C for 72 h. For further validation and to evaluate their survival curve, the bacterial isolates from radioactively contaminated plates were subcultured.

### 4.2. Identification of Radioresistant Bacterial Strain ASM-1

Strain ASM-1 isolated from the Makarwal region was selected based on its maximum survival rate under UVB exposure as well as its maximum antioxidant potential. The ASM-1 strain was identified morphologically as well as biochemically as per previously described approaches [26]. The molecular characterization was performed by employing 16S rRNA gene sequencing.

A DNA extraction kit (QIAGEN, Hilden Germany) was used to extract the DNA, and two universal primers were used to amplify the 16S rRNA gene sequence [27F′: AGAGTTTGATCMTGGCTCAG, 1492R′: TACGGYTACCTTGTTACGACTT] in PCR reactions. The Macrogen Service Center sequenced the amplified product (Geunchun-gu, Seoul, South Korea) and analyzed by a Nucleotide sequence alignment tool (nBLAST) available online in the National Center for Biotechnology Information (NCBI) database to ascertain the best matching genus. The ASM-1 strain, along with other homologs obtained from NCBI database, were subjected to computational analysis using Molecular Evolutionary Genetics Analysis (MEGA-X) to find out their phylogenetic relationships [27,28,29]. To identify the ASM-1 strain and analyzing the diversity of UV-resistant extremophiles, a neighbor-joining tree was built. To receive an accession number, the acquired sequence was uploaded to the NCBI GenBank.

### 4.3. Survival Curve of Strain ASM-1 at UVB and Oxidative Stress

After being exposed to various UVB radiation dosages, the ASM-1 strain’s capacity to survive was assessed using the procedure as described earlier by (Mattimore and Battista 1996) and the survival curve was plotted. Cells of strain ASM-1 were serially diluted (1:1000), using phosphate-buffered saline (PBS), and spread on tryptone glucose yeast (TGY) agar plates. The plates were exposed to various doses of 280 nm UV radiation and then incubated at 37 °C for 72 h. The rate of survival was calculated by dividing the total number of colonies on radioactive plates by the total number of colonies on non-radioactive plates [30]. Oxidative stress and Mitomycin-C tolerance were determined by diluting an overnight grown culture of the ASM-1 strain with sterile normal saline up to an optical density (OD600) 0.5, then treated with different molar concentrations (0–10 mmol L^−1^) of hydrogen peroxide (H_2_O_2_) for 30 min and mitomycin C (2–10 μg/mL) for 20 min, before inoculation on TGY plates. The number of colonies formed from the treated and untreated samples were compared after the plates were incubated at 37 °C for 72 h to evaluate the survival rate [25,31]. Each experiment was performed in triplicate.

### 4.4. Extraction of Extracellular Bioactive Compounds from Bacillus subtilis ASM-1 Strain

A total of 2.5 L of the overnight grown culture of strain ASM-1 in TGY broth was centrifuged at 10,000× *g* for 10 min at 4 °C and supernatant was collected. The extracellular metabolites were harvested from cell-free supernatant through liquid phase solvent extraction using an equal volume of ethyl acetate. The extracellular crude extracts (AS-2) were obtained in dried form using rotary evaporator.

### 4.5. Partial Purification of Extracellular Bioactive Compounds from B. subtilis ASM-1

The extracellular crude extract AS-2 from strain ASM-1 was subjected to purification through solid-phase extraction using HC-C18 SPE manual cartridge column (4.6 × 100 mm, 62 Å average pore size, 58 μm average particle size) with an average flow rate of 1.0 mL/min. The cartridge was loaded with 480 mg of crude extract (AS-2) and eluted using a series of solvents from polar to non-polar [water (W)—methanol (M)—ethyl acetate (E)—dichloromethane (D)], resulting in 4 different fractions in their respective solvents (Figure 8).

### 4.6. In Vitro Bioassays of Purified Fractions from B. subtilis ASM-1

#### 4.6.1. DPPH Radical Scavenging Assay

The antioxidant activity of selected purified fractions E-2 and D-2 was determined by 2,2-diphenyl-1-picrylhydrazyl (DPPH) radical scavenging assay [31]. A total of 100 μL of fractions was mixed in different concentrations (50, 100, 150, 200, and 250 µg/mL) with 100 μL of 0.1 mM of DPPH solution in microtiter plate and incubated at 37 °C for 30 min. Ascorbic acid was used as positive control while DPPH solution (200 μL) without a sample as negative control. After incubation in dark, the absorbance was read at 517 nm using an ELISA plate reader. The experiment was run in triplicates and the percent scavenging was calculated using Equation (2):% Scavenging = (Abs of control − Abs of sample)/(Abs of control) × 100(2)

#### 4.6.2. Hydrogen Peroxide Scavenging Assay

A total of 120 µL of purified fractions (E-2 and D-2) in different concentrations (100–250 µg/mL) was taken in a microtiter plate and then 80 µL of the hydrogen peroxide solution (40 mM) in 100 mM phosphate buffer (pH 7.4) was added into it [32]. The plates were incubated at 37 °C for 15 min and absorbance was measured at 230 nm using ELISA plate reader. A phosphate-buffered solution as a negative and ascorbic acid as a positive control was run separately. The experiment was run in triplicate and the scavenging percent was calculated using Equation (2).

#### 4.6.3. Cytotoxic and Antibacterial Potential of ASM-1 Fractions

The brine shrimp lethality assay was carried out to anticipate the cytotoxic potential of the extracts, with Mitomycin-C used as a standard by the method previously described in [33]. The antibacterial potential of the test extracts was investigated using a well diffusion assay against a range of different bacterial species, i.e., *Bacillus altitudinis* ASM-9, *E. coli* (ATCC 10536), *Pseudomonas aeruginosa*, *Listeria monocytogenes*, *Klebsiella pneumoniae*, *Staphylococcus aureus*, and *Staphylococcus epidermidis* (clinical strains).

#### 4.6.4. Biofilm Detection through Microtiter Plate Method

Both *Bacillus subtilis* ASM-1 and *Bacillus altitudinis* ASM-9 strains were evaluated for their biofilm formation ability through the tissue culture plate method as described by Christensen et al. [34]. A total of 200 µL of freshly grown cultures of ASM-1 and ASM-9 in LB broth with 1% glucose were inoculated in sterile 96 well plate and incubated at 37 °C for 72 h. Negative control wells contained uninoculated sterile broth. After incubation, the plates were washed and stained with 125 μL of 1% (*w*/*v*) crystal violet for 15 min. Absorbance of the remaining dye was measured at λ540 nm using an ELISA Microplate reader.

#### 4.6.5. Gas Chromatography–Mass Spectrometry Analysis for ASM-1 Fractions

The chemical constituents in most promising fractions (E-2 and D-2) with maximum antioxidant activity were analyzed by gas chromatography–mass spectrometry (GC–MS) (Agilent, Model number 6890 N) [35]. Agilent JW Scientific DB-5 MS capillary column (dimensions 30 m × 0.25 mm × 0.25 µm) fused with 5% phenyl methyl polysiloxane; Helium was used as carrier gas at a flow rate of 1 mL/min. The column temperature was initially set at 70 °C for 2 min followed by an increase of 3 °C/min to attain 250 °C for the total sample run time of 30 min. The ionization voltage for MS was operated at 70 eV with an ionization current of 0–315 µA. The instrument was set to detect compounds in a mass range of 50–1000 a.m.u. The chemical compounds were identified after comparing the mass spectral data with the National Institute of Standards and Technology (NIST) Ver. 02 library.

### 4.7. In Vitro Analysis of ASM-1 Extracts in Protecting Radiosensitive Microorganisms under UVB Radiation

The radioprotective property of extract from ASM-1 was analyzed against radiosensitive microorganisms under UV radiation stress. Two radiosensitive bacterial strains ASM-9 and *E. coli* (ATCC 10536) were spread on TGY agar plates (90 mm) and followed by spreading of 100 µL of liquid mixture containing E-2 fraction (1 mg/mL) and supernatant (24-hour-old culture) of a radioresistant ASM-1 strain in a ratio of (2:1). The plates were kept in a UVB chamber for UVB doses in exponential form for different time intervals. Thereafter, the plates were wrapped in aluminum foils and incubated for 3 days at 37 °C. A control with radiosensitive strains without spreading ASM-1 extracts was run in parallel as a control. The results were recorded after 3 days using the following formula and the data is expressed as a mean value obtained from triplicates as shown in Figure 9.
(3)$$\%Survival=\frac{No\;of\;colonies\;on\;UV\;irradiated\;plates}{No\;of\;colonies\;on\;control\;plates}\times 100$$

### 4.8. Ligand Preparation

The compounds were studied based on literature search where two compounds were obtained having role in inhibiting UV radiation among the compounds obtained via GC/MS analysis. These compounds are Pyrrolo [1,2-a] pyrazine -1,4-dione, hexahydro-3-(2- methyl propyl and 9H-Pyrido [3,4-b] indole. Ligand preprocessing, which included protonation, ionization, and the addition of explicit counter ions, hydrogen atoms, or atomic partial charges, was performed using Discovery Studio (DS) and UCSF Chimera. Energy minimization of small molecules was carried out using the AMBER ff14SB forcefield. The refined dataset obtained from these steps was then utilized for further computational experiments. To facilitate the study, ChemDraw (Mendelsohn, 2004), a chemical drawing tool, was used to draw the 2D structures of the compounds and convert them into 3D structures. Subsequently, ligand preprocessing was performed on these compounds.

### 4.9. Molecular Docking

Molecular docking of the compounds was carried out using the PyRx interface [36] and GOLD software [9]. For the docking process, the following parameters were employed: an exhaustiveness value of 10, the binding site was defined as the residues of the N-terminal domain involved in the binding pocket, and a maximum of 10 poses were allowed. The PyRx tool generated binding affinity values in the negative (where a more negative value indicates a stronger binding affinity). The obtained inhibitors demonstrated a high binding affinity were selected for further analysis. The docking results of the top inhibitors can be found in Figure 5 and Figure 6. To validate the docking protocol used, an inhibitor was docked into the N-terminal structure of (PDB ID = 51FA and 5KZP) using 10 iterations. The same binding mode as the one observed in crystallization studies for the known inhibitors was obtained. The docking reproducibility results obtained using AutoDock Vina are presented in Figure 5 and Figure 6.

### 4.10. Molecular Dynamics Simulations

The docked solutions and inhibitor were subjected to a 20 ns molecular dynamics (MD) simulation using AMBER16 [37]. The inhibitor was optimized with the AMBER (GAFF) force field [38], while the targets protein parameters were generated using the ff14SB force field [39]. To integrate the complex into a TIP3P water box, a padding distance of 12 between target proteins and the box borders was set. Sodium ions were added to neutralize the system. The system was heated to 300 K (NVT) using Langevin dynamics [40] for 20 ps to maintain a constant temperature [41]. A time step of 2 fs was used, and a 5 kcal/mol-Å^2^ restriction was applied to carbon alpha atoms. The system was relaxed for 100 milliseconds during equilibration. To maintain system pressure, a 50 ps NPT ensemble was employed. Finally, a 20 ns production run was conducted at a 2 fs rate. The trajectories generated were analyzed for structural parameters using the AMBER CPPTRAJ program [42]. The hydrogen bonds formed between protein targets and the inhibitor throughout the trajectories were visualized.

## 5. Conclusions

In summary, this study demonstrates the isolation of a radioresistant bacterial strain from the Thal desert, Pakistan. The strain was identified as *Bacillus subtilis* strain ASM-1 (OK559666) and was able to tolerate the detrimental effects of UVB radiation and Mitomycin-C up to a significant level. The extracellular ethyl acetate fraction as compared to other fractions possessed considerable antioxidant potential evaluated via DPPH and H_2_O_2_ scavenging assays. The chemical profile of the partially purified fractions concluded from GC–MS spectra was revealed to have phenolic compounds, pyrrolopyrazine, and other important constituents having significant antioxidant and radioprotective potential. Consequently, ASM-1 unveiled its hidden potential of having a radioprotective role to aid in the survival quest of radiosensitive microorganisms under UVB radiation. Thus, the present study concluded the antioxidative, and radioprotective potential of *Bacillus subtilis* strain ASM-1 extracts, which could play a crucial role in the bioremediation studies of radioactive wastes via growth cooperation with the efficient radiosensitive microbial degraders.

## Figures and Tables

**Figure 1 pharmaceuticals-16-01139-f001:**
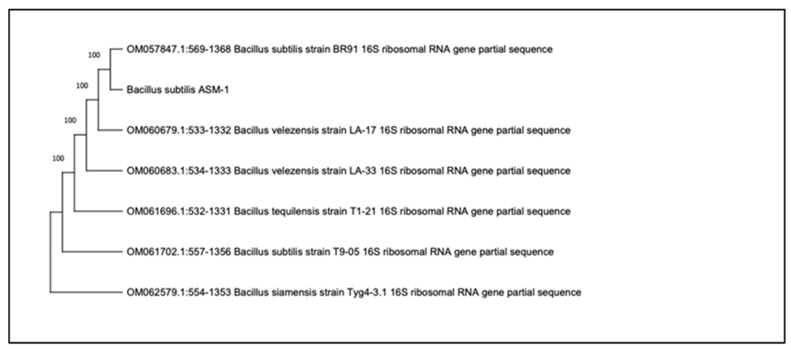
The phylogenetic tree was constructed for ASM-1 using the Neighbor-Joining approach using Molecular Evolutionary Genetics Analysis (MEGA-X) software with a bootstrap value (1000 replicates) of 100 to *Bacillus subtilis* strain ASM-1(OK559666).

**Figure 2 pharmaceuticals-16-01139-f002:**
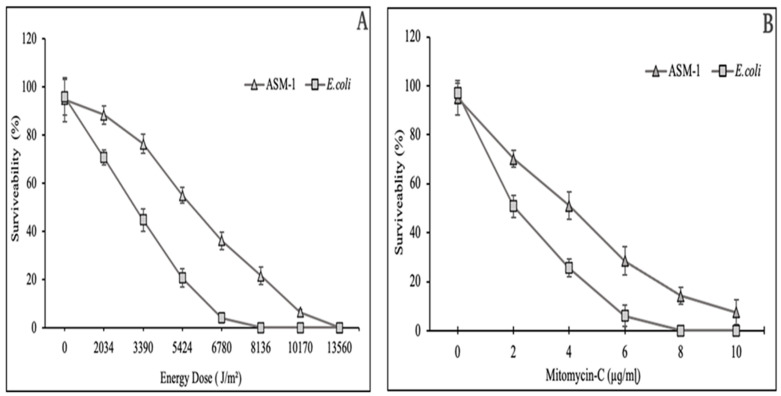
Survivability rate determination for *Bacillus subtilis* strain ASM-1 under UVB radiation (**A**), and Mitomycin-C (**B**). The data is calculated and expressed from the average of three replicates ± SDs.

**Figure 3 pharmaceuticals-16-01139-f003:**
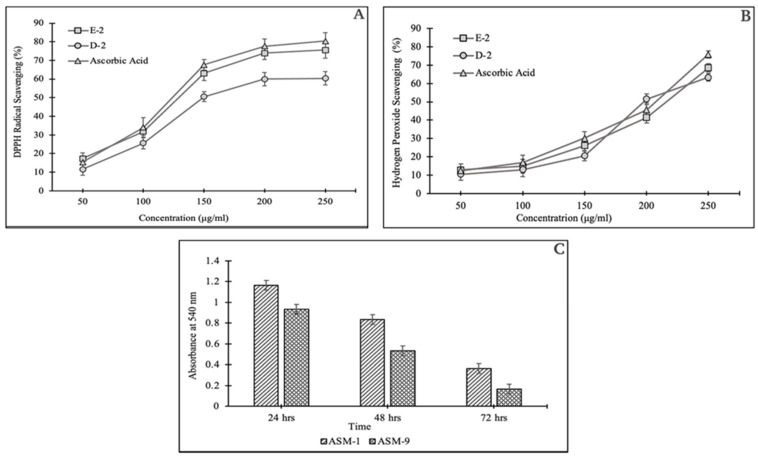
The selected fractions obtained from ASM-1 extracts were evaluated for their antioxidant potential via DPPH free-radical scavenging assay (**A**), hydrogen peroxide assay (**B**). Similarly, the biofilm formation potential for both isolates (ASM-1, and ASM-9) are depicted in the bar graph (**C**). The data expressed here was calculated as a mean value obtained from the triplicates showing ± SDs.

**Figure 4 pharmaceuticals-16-01139-f004:**
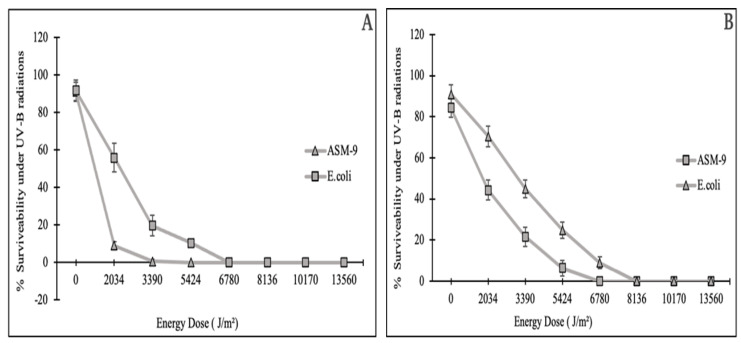
Growth and survivability rate of radiosensitive strains (*E. coli* and ASM-9) under UVB radiation with no ASM-1 extracts (**A**). The second figure shows survivability of radiosensitive strains after the addition of ASM-1 extracts (**B**). The data are expressed as mean values of the triplicates ± SDs.

**Figure 5 pharmaceuticals-16-01139-f005:**
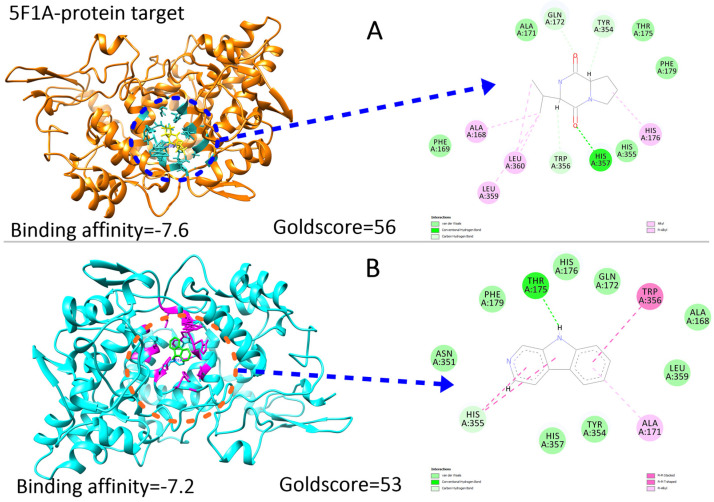
Depicting the protein target with binding interactions of both compounds against the target 5F1A with binding affinities and Goldscore. (**A**,**B**) Presenting the active cavity of the protein where both ligand molecules are attached as shown in blue and orange circle on left whereas 2D image of both (**A**,**B**) shows the binding residues of the active pocket especially hydrogen bonding.

**Figure 6 pharmaceuticals-16-01139-f006:**
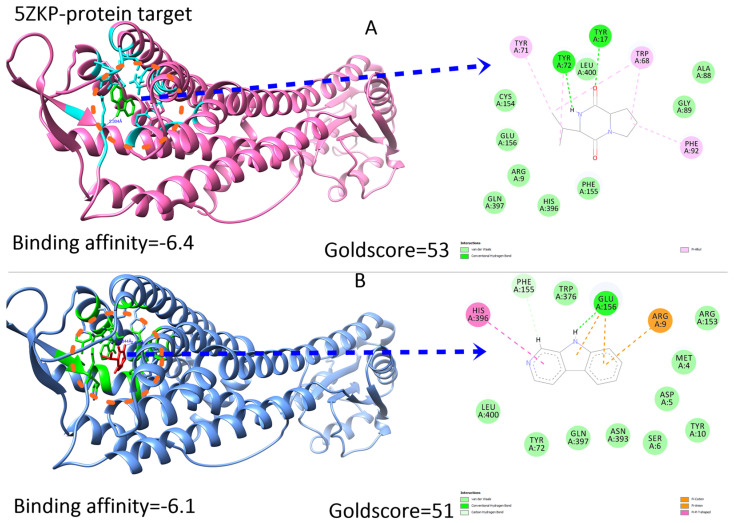
Depicting the protein target with binding interactions of both compounds against the target 5ZKP with binding affinities and Goldscore. Both (**A**,**B**) show the binding cavity where both inhibitors are docked to specific active residues as shown in orange circle on left side. Whereas the 2D image on right side presenting the hydrogen bonds, van der waals and other interactions holding the ligand inside the pocket.

**Figure 7 pharmaceuticals-16-01139-f007:**
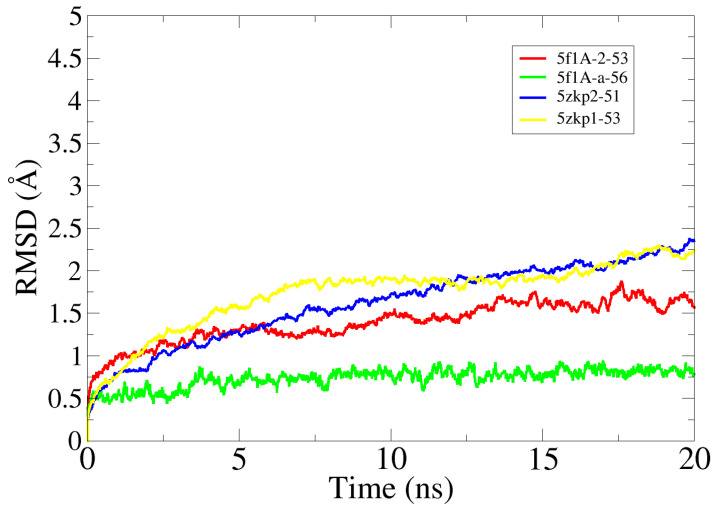
RMSD analysis of simulation complexes based on carbon alpha atoms.

**Figure 8 pharmaceuticals-16-01139-f008:**
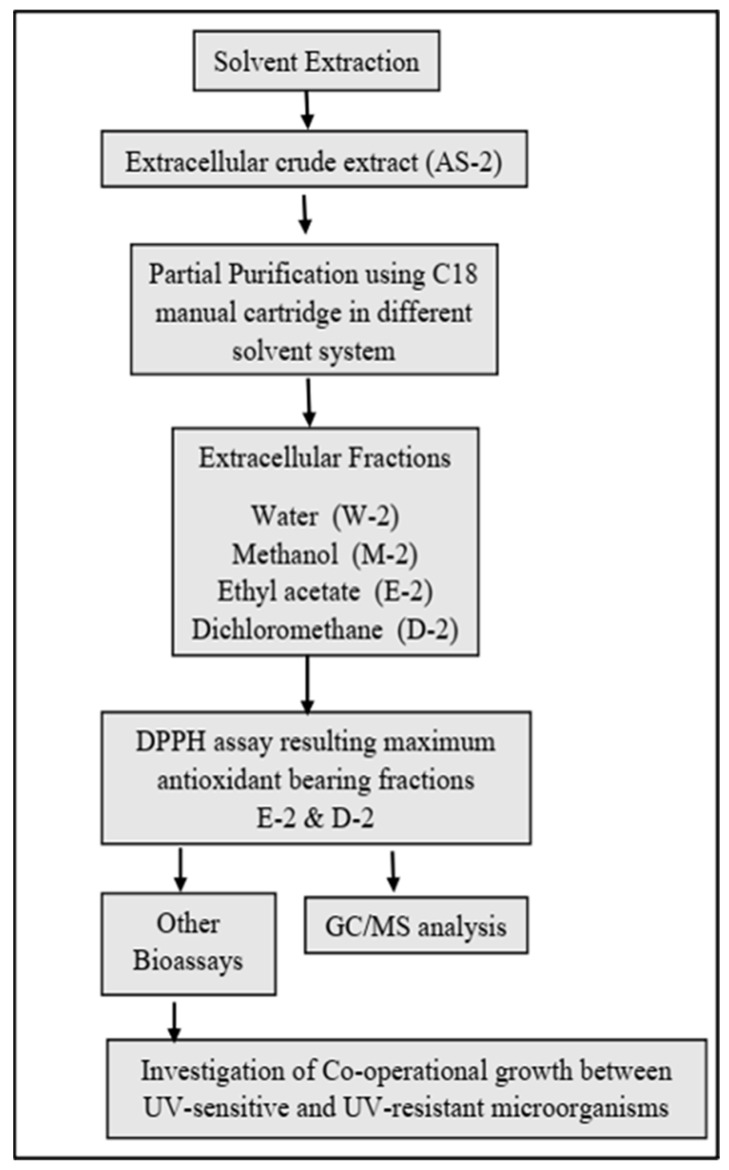
A schematic representation of the stepwise extraction, fractionation, and evaluation process for the extracellular crude extracts obtained from strain ASM-1.

**Figure 9 pharmaceuticals-16-01139-f009:**
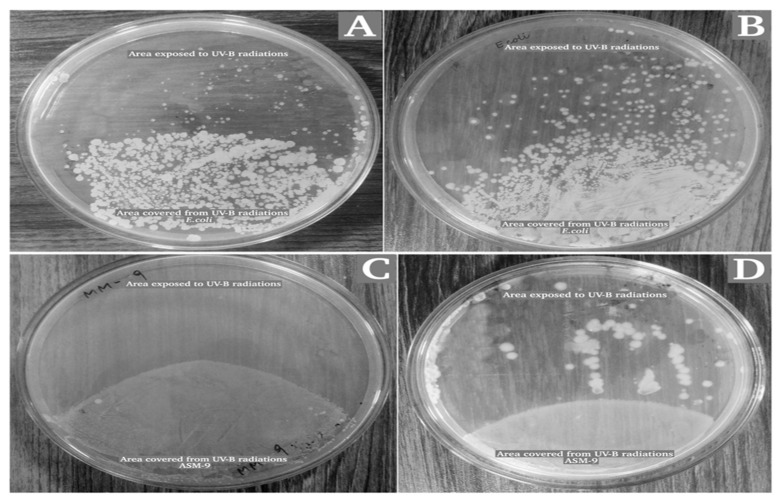
Growth and survivability rate of radiosensitive strains (*E. coli* and ASM-9) under UVB radiation with no ASM-1 extracts (**A**). The second figure shows survivability of radiosensitive strains after the addition of ASM-1 extracts (**B**). The data are expressed as mean values of the triplicates ± SDs. (**C**) Depicts the survival of ASM 9 under UV radiation without the addition of ASM 1 extracts. Whereas, (**D**) shows the survival of ASM 9 under UV radiation after the addition of ASM 1 extracts.

**Table 1 pharmaceuticals-16-01139-t001:** Primary and secondary screening for radioresistant isolates with radiation exposure time and energy dose.

Exposure Time	Radiant Exposure in (J/m^2^) He = Ee × t (Sec)	UV Resistant Isolates
3 min	2.034 × 10^3^ J/m^2^	* TMC 1 to TMC 18 ** ASM 1 to ASM 8
5 min	3.390 × 10^3^ J/m^2^	TMC 2, TMC 7, TMC 6, TMC 9, TMC 11, TMC 12, TMC15, ASM 2, ASM 1, ASM 3, ASM 5, TMM 6, ASM 8
8 min	5.424 × 10^3^ J/m^2^	TMC 2, TMC 7, TMC 6, TMC 9, TMC 12, TMC15, ASM 1, ASM 5, ASM 6, ASM 8
10 min	6.780 × 10^3^ J/m^2^	ASM-1, ASM-6, ASM-8
12 min	8.136 × 10^3^ J/m^2^	ASM-1

* Chashma sample denoted by TMC; ** Makarwal sample denoted by ASM.

**Table 2 pharmaceuticals-16-01139-t002:** Major constituents detected in the GC–MS spectra for the selected fractions of ASM-1 extracts.

Fractions	Retention Time	Content (%)	Chemical Compounds	Molecular Formula	Molecular Weight	Class
E-2	3.083	1.03	Pyrrolo [1,2-a] pyrazine -1,4-dione, hexahydro-3-(2-methyl propyl)	C_11_H_18_N_2_O_2_	210	Pyrrolopyrazine
	4.146	0.14	9H-Pyrido [3,4-b] indole	C_11_H_8_N_2_	168	β-carboline alkaloid
	4.123	0.56	Pyrrolo [1,2a] pyrazine-1,4-dione, hexahydro -3-(phenylmethyl)	C_7_H_10_N_2_O_2_	154	Pyrrolopyrazine
	7.521	0.77	n-Nonadecanol-1	C_19_H_40_O	284	Alcohol
	11.614	0.21	Dodecane	C_12_H_26_	170	Alkane
	15.291	1.91	Phenol,2,2′-methylenebis [6-(1,1-dimethylethyl)-4-methyl	C_23_H_32_O_2_	340	Phenol
D-2	3.412	1.09	Benzamide	C_7_H_7_NO	121	Amide
	4.243	0.28	1,2-benzenedicarboxylic acid, monobutyl ester	C_12_H_14_O_4_	222	Dicarboxylic acid Butyl ester
	5.461	0.32	1,2-Benzenedicarboxylic acid, diisooctyl ester	C_24_H_38_O_4_	390	Dicarboxylic acid Octyl ester
	11.065	0.37	(S)-6-Hydroxyheptanoic acid	C_7_H_14_O_2_	130	Fatty acid
	11.534	0.04	1-Octanol	C8H18O	130	Alcohol

## Data Availability

Data is contained within the article.

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
