# Peer review of "Unraveling the Radioprotective Mechanisms of UV-Resistant Bacillus subtilis ASM-1 Extracted Compounds through Molecular Docking"

_pharmaceuticals, 2023, doi:10.3390/ph16081139_

Round 1

Reviewer 1 Report

Presented studies results are very interesting. Manuscript is written clearly and sections are well planned by authors.

One topic is not clear for me. Table 2 "Major constituents detected in the GC-MS spectra for the selected fractions of ASM-1 ex- 167 tracts". Some of described compounds e.g. pyrrolopyrazine possess carcinogenic properties. Cytotoxicity was assessed, but not in terms of found residues. Please comment.

Lack of English issues.

Author Response

Comments and Suggestions for Authors

Presented studies results are very interesting. Manuscript is written clearly and sections are well planned by authors.

One topic is not clear for me. Table 2 "Major constituents detected in the GC-MS spectra for the selected fractions of ASM-1 ex- 167 tracts". Some of described compounds e.g. pyrrolopyrazine possess carcinogenic properties. Cytotoxicity was assessed, but not in terms of found residues. Please comment.

Response

Thank you for highlighting this. Yes, this is a valid question. That the presence of pyrrolopyrazine compounds with known cytotoxic potential in the GC-MS spectra but lack of its cytotoxicity in the brine shrimp lethality assay could be attributable to several reasons, including:

  1. Concentration: The concentration of pyrrolopyrazine in the partially purified crude extracts may be too low to elicit a cytotoxic reaction in the brine shrimp experiment. The substance could be present in trace amounts or at concentrations below the threshold necessary for observable cytotoxic effects.

  1. Synergistic Effects: The pyrrolopyrazine was present in the crude extract alongside other compounds that may have altered its biological activity. Because some chemicals have a synergistic impact, it is possible that it decreased the pyrrolopyrazine's cytotoxic potential. Similarly, taking interference into account, other metabolites present in the crude extract may have interfered in the cytotoxicity experiment, resulting to false-negative results in this scenario.

  1. Bioavailability: A third possibility is that the pyrrolopyrazine found in the GC-MS spectra might not be readily bioavailable or bioaccessible to brine shrimp larvae, resulting in a lack of detectable cytotoxicity.

  1. Selective toxicity: The pyrrolopyrazine may be cytotoxic to some cell types but might not to the brine shrimp larvae utilized in the assay. Because most compounds have selective toxicity i.e., It can have different effects on different cell lines or organisms.

Reviewer 2 Report

In this work, the authors studied ASM-1 and showed its potential of playing a radioprotective role for microorganisms survival under UVB-radiation. While the experimental part looks good to me, my major concern is about the molecular docking part in this work. I understand the goal of using docking in this work is to understand the binding mode and binding affinity of studied compounds. However, previous studies have shown docking is good for binding mode prediction but the predicted binding affinities do not correlate with the experimental measured values. First, a docking score cannot be considered as a binding affinity since several approximations are implemented in the simplified docking scores. Many important degrees of freedom describing entropic contributions of the ligand and protein are omitted (e.g., protein reorganization and desolvation effects). So in theory a docking score cannot represent the true binding affinity even though the docking algorithms use an energy like unit (kcal/mol). Previous studies have shown how bad the docking scores correlate with the experimental binding affinities (see https://doi.org/10.1002/cmdc.202200425,  https://doi.org/10.1021/jm050362n, https://doi.org/10.3390/molecules23081899). That said, the authors should highlight the limitation of using docking scores to estimate binding affinities of these studied compounds with these previous studies in the manuscript so that the readers are not biased. Alternatively, the authors can also provide experimental measured binding affinities to validate these predictions. Also, even though docking has shown great success in predicting binding poses, I suggest the authors perform short molecular dynamics simulations to validate these predicted poses and interactions. If the authors do not have the resource to perform these simulations it should be noted in the manuscript so that the readers are aware of the potential limitations of these results.

Author Response

Comments and Suggestions for Authors

In this work, the authors studied ASM-1 and showed its potential of playing a radioprotective role for microorganisms survival under UVB-radiation. While the experimental part looks good to me, my major concern is about the molecular docking part in this work. I understand the goal of using docking in this work is to understand the binding mode and binding affinity of studied compounds. However, previous studies have shown docking is good for binding mode prediction but the predicted binding affinities do not correlate with the experimental measured values. First, a docking score cannot be considered as a binding affinity since several approximations are implemented in the simplified docking scores. Many important degrees of freedom describing entropic contributions of the ligand and protein are omitted (e.g., protein reorganization and desolvation effects). So in theory a docking score cannot represent the true binding affinity even though the docking algorithms use an energy like unit (kcal/mol). Previous studies have shown how bad the docking scores correlate with the experimental binding affinities (see https://doi.org/10.1002/cmdc.202200425,  https://doi.org/10.1021/jm050362n, https://doi.org/10.3390/molecules23081899). That said, the authors should highlight the limitation of using docking scores to estimate binding affinities of these studied compounds with these previous studies in the manuscript so that the readers are not biased. Alternatively, the authors can also provide experimental measured binding affinities to validate these predictions. Also, even though docking has shown great success in predicting binding poses, I suggest the authors perform short molecular dynamics simulations to validate these predicted poses and interactions. If the authors do not have the resource to perform these simulations it should be noted in the manuscript so that the readers are aware of the potential limitations of these results.

Response

We are thankful to the reviewer for highlighting the issues and suggestions. Yes, we agree upon the validation of docking poses and the correlation between binding affinities and experimental measured values, thus we have performed molecular dynamics study to check dynamics of the whole complex systems as suggested by the reviewer in the revised manuscript at page 10-11 at line 239-253.
